# Velocity-Based Training in Soccer: A Brief Narrative Review with Practical Recommendations

**DOI:** 10.3390/jfmk10020106

**Published:** 2025-03-26

**Authors:** Andrés Rojas Jaramillo, Richard B. Kreider, Jorge L. Petro, Diego A. Bonilla, Juan José Gonzalez-Badillo, David Rodriguez-Rosell

**Affiliations:** 1Educational and Pedagogical Studies and Research Group (GEIEP), Faculty of Education, Department in Physical Education, Corporación Universitaria Minuto de Dios, Bello 050001, Colombia; 2Exercise & Sport Nutrition Laboratory, Human Clinical Research Facility, Texas A&M University, College Station, TX 77843, USA; rbkreider@tamu.edu; 3Research Division, Dynamical Business & Science Society—DBSS International SAS, Bogotá 110311, Colombia; jlpetro@dbss.pro (J.L.P.); dabonilla@dbss.pro (D.A.B.); 4Research Group in Physical Activity, Sports and Health Sciences-GICAFS, Universidad de Córdoba, Montería 230002, Colombia; 5Physical Performance & Sports Research Center, Universidad Pablo de Olavide, 41013 Seville, Spain; jjgbadi@gmail.com (J.J.G.-B.); drodros@upo.es (D.R.-R.); 6Department of Sport and Informatics, Universidad Pablo de Olavide, 41013 Seville, Spain; 7Investigation in Medicine and Sport Department, Research, Development and Innovation (R&D+i) Area, Sevilla Football Club, 41005 Sevilla, Spain

**Keywords:** strength training, soccer players, resistance exercises, determinant factor, physical improvements

## Abstract

Soccer is an intermittent sport characterized by periods of high intensity interspersed with varying levels of recovery. During the game, soccer players are required to perform numerous determinant actions, such as accelerations, decelerations, pace and directional changes, jumps, and strikes, which demand high levels of strength, speed, endurance, and mobility. This review aims to examine the current scientific evidence on velocity-based training (VBT) in order to assess its role in optimizing key actions in soccer players and to provide practical applications for its implementation. A narrative review of the scientific literature was conducted, focusing on velocity-based resistance training and its application in soccer, identifying relevant studies on its effectiveness in improving specific physical capacities and sports performance. Velocity-based resistance training is an effective strategy for enhancing key skills such as linear sprints, vertical jumps, and direction changes. Additionally, this methodology allows for the real-time adjustment of training loads, optimizing outcomes and minimizing muscle fatigue. Velocity-based resistance training represents an innovative and practical tool for soccer coaches and strength and conditioning professionals. Its application can contribute to optimizing physical performance, reducing injury risks, and improving sport-specific capabilities.

## 1. Introduction

Enhancing performance in team sports, including soccer, requires the improvement of conditional or physical capacities such as strength, endurance, and speed [1,2,3,4,5,6,7]. For this, it is necessary to implement training programs focused on the simultaneous development of the different physical variables [8,9,10,11,12,13]. Despite the recognized importance of physical condition, specifically muscle strength, for performance in soccer players, a consensus remains elusive among coaches and strength and conditioning professionals regarding the most efficient training methodologies and their optimal integration within the broader training regimen for soccer players [8,14].

When discussing physical preparation, reference is made to four fundamental conditional capacities: strength, endurance, speed, and mobility [15]. In the case of endurance training, multiple methodologies exist, ranging from continuous training to interval methods or even modified game situations [16]. However, there is no clear consensus on the best approach, as its application depends on the specific demands of each sport. This aspect will be analyzed in Section 2, which focuses on physical preparation in soccer.

On the other hand, strength training is considered one of the basic pillars of training and, consequently, has been widely studied in soccer players [15]. Over time, several methodologies have been proposed, including highly specific exercises [17], bodyweight exercises [18], and heavy-load training [18]. However, one of the main limitations of any proposed training method is the lack of precise and objective training load control. This issue arose because, until relatively recently, there was no valid, precise, and objective method to quantify training load. The solution to this challenge is the implementation of velocity-based training [19,20].

Using this procedure (i.e., velocity as a means of monitoring relative load and volume during resistance training), numerous studies were conducted comparing different volumes [21], relative loads [22,23], and loading parameters, improving the understanding of sport-specific adaptations and identifying the minimum effective dose required to achieve optimal benefits [21]. Furthermore, these studies have highlighted the influence of resistance training on key performance actions, such as sprint speed over different distances and jump and change of direction ability [18].

However, the use of velocity as a means for monitoring resistance training is often misunderstood and, in many cases, inaccurately described in the literature [20]. This has hindered its application and comprehension among coaches, who frequently perceive it as impractical or difficult to implement. Therefore, this article aims to clarify the practical application of this methodology and present the latest findings from the scientific literature on the subject. Thus, this review aims to examine the current scientific evidence on velocity-based training (VBT) in order to assess its role in optimizing key actions in soccer players and to provide practical applications for its implementation.

## 2. Physical Preparation in Soccer

Soccer is often classified as an endurance-based sport due to an average match intensity of ~80% HRmax and distances covered between 9 and 13 km [24] (Table 1). Research suggests that the total match distance remains largely consistent across leagues and competitive levels, questioning its role as a key performance determinant [25,26].

Although Table 1 provides relevant data, it appears that the variable total match distance alone does not define physical performance in soccer players. A deeper analysis of in-game actions is crucial to identify key performance factors, especially those frequently executed or linked to decisive plays. This approach helps clarify the actual impact of distance covered on player performance.

Several studies have focused on analyzing players’ activity patterns during a match. Most of these studies reveal that 61.6% of the total distance covered occurs at speeds below 11 km·h^−1^ [2,4,27,28,29,30]. This observation suggests that a significant portion of match time is spent walking or engaging in low to very low intensity running [2,30]. On the other hand, ~30% of the total distance covered occurs at moderate speed (from 11 km·h^−1^ to 19 km·h^−1^) and the remaining 10% is covered at a high or maximum speed [2,3,4,27,28,30]. In fact, only 3–4% of the total distance is covered at speeds above 24 km·h^−1^ [2,27,30,31]. In addition, while various studies have shown that the average speed during the match has progressively increased over several seasons, the total distance covered has remained similar or decreased slightly [32,33,34]. Therefore, based on these data, it could be inferred that the total distance covered during a match does not appear to be a decisive performance determinant in soccer, nor does it suffice as a valid indicator for assessing a player’s fitness level or performance capacity in professional soccer players.

On the other hand, despite the fact that less than 15% of the total distance covered during gameplay is performed at high or maximal intensity [2,28,30], some authors have posited that these actions hold paramount importance within the context of the game, as they heighten the likelihood of scoring or preventing a goal in both male [35,36] and female [37] soccer players. Consequently, several studies [31,38] have decided to focus on high-speed running (HSR) and the distance covered at maximum speeds (sprints) because it seems that these variables have more relation to soccer performance than the total distance covered.

An analysis to identify the determinants of physical performance in soccer could be based on the examination of player actions leading up to scoring or assisting, as these actions notably influence match outcomes. In this regard, a study [35] analyzed the prevalent actions in such scenarios by examining 360 goal sequences and 322 assists from the German Professional League’s 2006 season in male soccer players. The most recurrent action before scoring, undertaken by both scoring and assisting players, was a linear sprint, followed by a vertical jump (scoring player) or a turn (assisting player), or a sprint with change in direction [35]. Specifically, linear sprints are observed in 61% of the instances preceding a goal, predominantly executed without ball possession. Moreover, a jump is performed before the goal in 22% of these instances, all occurring without ball engagement. Regarding actions by assisting players, linear sprints are performed 67% of the time, primarily with ball possession. Additionally, rotational body movements account for 13% of instances, jumps for 11%, and changes in directions for 9% [35]. Similarly, a recent investigation also with male soccer players [36] analyzed all goals in the English Premier League throughout the 2018–2019 season. Again, findings revealed that the most frequent individual movement preceding a goal was a linear sprint (mean ± SD, 32.4 ± 1.3%), followed by deceleration (20.2 ± 0.9%) and turns (19.8 ± 0.9%). It is worth noting that in 82.9 ± 1.5% of player involvements, at least one high-intensity movement was observed [36]. Similarly to male soccer players, data from all the goals on the Women’s Super League 2018/2019 were collected [37]. The results showed that linear advancing motion (mainly sprint actions) (total percentage [95% CI] 37% attackers and 32.7% defenders) was the most common action preceding a goal, followed by deceleration (21.5% attackers; 18.4% defenders) and turn (19.2% attackers; 17.6% defenders) [37].

These studies [35,36,37] offer valuable insights into the specific goals for the physical conditioning of professional soccer players, particularly regarding training programs suitable for these athletes. It seems that the primary focus should be on enhancing key performance indicators such as linear sprinting, vertical jumping, rotational movements, pivoting, change in direction, and repeated sprints [2,3,4,39]. All these actions crucially rely on the rapid manifestation of force by the lower limb musculature, specifically emphasizing high force output per unit of time [40]. This concept is generally known as the rate of force development (RFD) [41,42].

In addition to the above, the significance of high-intensity and high-speed movements, particularly linear sprinting, is underscored by the evolving metrics of distance covered at high or maximum speeds during competitive matches. Barnes et al. [32] conducted a longitudinal analysis spanning seven consecutive seasons (from 2006 to 2007 to 2012–2013) across Premier League teams, examining the evolution in distances covered at various speed ranges. Despite the overall average total distance covered by players remaining relatively stable over the study period (10–12 km), there was a progressive increase (~30–35%) in both HSR and sprinting distances throughout the seasons under analysis [32]. In this line, other studies also analyzing the evolution of physical performance variables depending on the position in the Chinese Soccer Super League [34] and Spanish *LaLiga* [33] have reported similar results. More recently, a study [43] examined the physical demands of teams during the FIFA World Cup Qatar 2022, contrasting them with those of the FIFA World Cup Russia 2018. This analysis revealed a mere 3% increase in total distance covered by national teams, contrasted with a substantial rise of 16–19% in distances covered at the higher intensities [43]. Furthermore, an additional dimension spotlighting the importance of high-intensity actions in match performance emerges from a study involving 97 soccer players who transitioned from competing in the Spanish second division to the LaLiga EA Sports (first division) [44]. These players exhibited a greater number of high-intensity running efforts and greater distance covered at high-intensity running when playing in the first division compared to their performance in the second division [44].

These findings suggest that match activity patterns are shaped by increased action speed, highlighting the need to prioritize this aspect in training. Therefore, to optimize performance in competitive soccer, physical preparation should focus on improving (Figure 1) the following:Linear sprint capacity;Vertical jump ability;Kicking ball speed;Sprinting with directional changes;Repeated sprint ability with short recovery.

In summary, training for soccer players should prioritize increasing applied force in competitive movements and maintaining high force levels as movement velocity rises, especially in the lower limbs. Once key performance actions and training objectives are identified, it is essential to refine the training approach by addressing critical questions on conditioning development.

Despite the growing implementation of VBT in soccer, there is still no clear consensus on its optimal application within physical preparation programs. While traditional strength training methodologies remain widely used, recent research highlights the importance of monitoring movement velocity to optimize neuromuscular adaptations, manage fatigue, and individualize training loads. However, the integration of VBT into soccer training lacks a structured approach, and many practitioners struggle with its application due to a lack of knowledge about its practical implementation. This has created a gap between scientific evidence and field application. Therefore, this review not only aims to analyze key studies and the relevance of VBT in soccer performance but also to bridge the gap between research and practice, providing concrete recommendations for coaches and strength and conditioning professionals to facilitate its implementation.

## 3. Methodology

This narrative review aims to synthesize existing knowledge on VBT and its application in soccer. Unlike a systematic review, this approach allows for a more flexible and critical discussion of the relevant literature without strictly predefined inclusion and exclusion criteria.

### 3.1. Study Selection Criteria

We included studies that have had a significant impact on the field of VBT, with a particular focus on the following:Research applying VBT in soccer and other sport modalities, considering the specificity of velocity and the type of exercise used.Studies designed to evaluate validity, reliability, or accuracy of load control through movement velocity.Interventions aimed at improving key physical capacities in soccer, such as sprinting speed, rate of force development (RFD), and change of direction ability.Systematic or narrative reviews that provide conceptual and empirical support for using the velocity of the first repetition and velocity loss within a set as strategies for training load control.

### 3.2. Exclusion Criteria

To avoid misinterpretations of the VBT concept, we excluded the following:Studies that divide training into velocity zones, as this approach misrepresents the original concept of velocity-based load control.Research that considers VBT as a training methodology in itself, rather than a tool or procedure for quantifying and controlling resistance training. This misconception suggests a lack of understanding of the true concept of VBT.

### 3.3. Sources and Databases

The literature search was conducted using high-impact scientific databases, including the following:PubMed;Scopus;Web of Science;SportDiscus.

Additionally, secondary references from key articles were reviewed to ensure broader coverage of the topic.

### 3.4. Search Period

We included studies published from 2010 to the present, prioritizing recent research that reflects the latest advancements in the application of VBT in soccer.

### 3.5. Keywords Used

The search was conducted using combinations of English terms, such as the following:“velocity-based training” AND “soccer”;“strength training” AND “velocity monitoring”;“load control” AND “velocity loss” AND “soccer players”;“velocity of first repetition” AND “fatigue monitoring”.

### 3.6. Analysis Strategy

The selected studies were analyzed and categorized based on their contribution to VBT knowledge in soccer. They were grouped according to the following:Evidence on the relationship between velocity and training load.Applications of VBT in soccer and its impact on physical performance.Methods for monitoring fatigue through execution velocity.Limitations and challenges in the implementation of VBT in sports settings.

### 3.7. Methodological Limitations

Since this is a narrative review, a systematic study selection protocol was not followed. This may introduce a selection bias, as the inclusion of articles was partially based on the authors’ criteria. However, this approach allowed for a deeper and more critical discussion of the available evidence, facilitating the formulation of practical recommendations based on the most relevant literature.

## 4. What Characteristics Should Strength Training Programs Have to Improve High-Velocity Musculoskeletal Actions?

All the previously mentioned determinant actions (i.e., linear sprint, vertical jump, kicking ball speed, sprinting with directional changes, and repeated sprint ability with short recovery) share a common characteristic: they depend on the ability to generate force per unit of time (RFD) [40]. Thus, enhancing RFD should be a primary goal of training programs for soccer players. For addressing this, a myriad of training methods has been used to increase the different manifestations of force, with resistance training (RT) being the most conducted in soccer players [8,45,46,47]. Traditionally, it was indicated that RT involving high relative loads (>70%) and repetitions carried out to or near muscular failure are requisite for achieving substantial gains in maximal strength and improvements in high-speed actions like jumping and sprinting [46,48,49,50,51,52,53,54]. However, recent evidence involving soccer, futsal, and basketball players across various age groups and competition levels [22,23,55,56,57,58,59,60] has shown that such training stimuli may not be imperative for performance enhancement. Indeed, training with relative loads ranging between 40 and 60% 1-RM in the back squat exercise, executing each repetition at maximum intended concentric velocity, and engaging in a low number of sets (e.g., 3–4) and repetitions per set (i.e., 4–6), combined with jumping and acceleration exercises, produces notable improvements in lower limb strength, jump and acceleration capacities, kicking ball speed, and the capacity to repeat sprints with short recovery times [22,23,55,56,57,58,59,60]. Additionally, RT involving sets performed to muscular failure induces a higher degree of fatigue, limiting subsequent field-specific training and interfering with players’ technical-tactical performance [61,62]. Consequently, it appears unnecessary to undertake RT with elevated levels of effort to achieve greater performance enhancements.

## 5. How to Quantify the Load or Degree of Fatigue During Strength Training?

Several variables can be manipulated to configure the training stimulus during RT, including relative load, volume, intra-set rest periods, sets, exercises, training sessions and frequency, density, execution velocity, and the type and order of exercises. While all these factors may influence outcomes, the degree of effort or fatigue induced by a specific resistance exercise appears primarily determined by the relative load and volume [63,64,65].

Traditionally, relative load in RT has been determined through percentages of one-repetition maximum (1-RM) [63,64,65,66,67]. However, this method faces some constraints, including challenges in administering the test, accurately determining 1-RM, and elevated risk of injury for participants,—particularly among young, older adults, and inexperienced subjects [19]. Also, the substantial time consumption, and notably, fluctuations in 1-RM values after several training sessions are factors that limit the use of this procedure for an accurate determination of the relative load [19]. Consequently, the programmed effort may not match the actual effort exerted by the athlete in successive training sessions. In the recent decade, it has been demonstrated that relative load during RT could be accurately and reliably determined through movement velocity (MV) [19,20,68,69]. This is due to the close relationship between the percentage of 1-RM displaced and MV, wherein each percentage of 1-RM corresponds to a determined MV value [20,68,69]. This statement is due to the fact that the load–velocity relationship remains constant regardless of the initial performance level of the individuals and the changes that occur in strength after a training period [68,69]. Hence, it is possible to accurately determine the relative load represented by a given weight by simply measuring the velocity at which it is moved during a maximal effort repetition [19,20]. Additionally, this approach allows for estimating the individual’s maximal lifting capacity (i.e., the 1-RM value) in an exercise without the need for a 1-RM test [19,20,69]. The assessment of MV can be conducted using various devices, with linear velocity transducers (LVTs) or linear position transducers (LPTs) showing superior validity and measurement agreement compared to accelerometry systems, video cameras, or mobile applications. Table 2 presents the velocities associated with different relative loads across various resistance exercises [19,68,69,70,71]. An important aspect to consider when prescribing relative load through MV (the so-called resistance training based on velocity control) is that the velocity associated with each relative load is specific to each exercise [19,20,68,69,70,71]. For example, the velocity associated with 60% 1-RM is 0.78 m·s^−1^ for bench press, 0.85 m·s^−1^ for pull-ups, 1.00 m·s^−1^ for back squat, and 1.06 m·s^−1^ for the bench rowing exercise (see Table 2).

Recent evidence suggests that the relationship between lifting velocity and repetitions to failure presents a strong fit and reliability, particularly in trained individuals [72]. This reinforces the application of velocity monitoring to guide training load and fatigue management in soccer players.

Alternatively, volume has conventionally been prescribed based on the number of sets and repetitions per set [63,73]. However, there is a notable inter-individual variability in the maximum repetitions achievable at a specific relative load [74,75]. Therefore, if two players perform an identical number of repetitions at a given relative load (e.g., 50% 1-RM), it is probable that they are experiencing different degrees of fatigue, as they will be completing differing proportions relative to their respective maximum achievable repetitions at that relative load [20,74,75].

To equalize effort levels across different individuals during RT, recent studies have demonstrated the utility of the percentage of velocity loss in the set (%VL) [20,74,75]. This is attributed to the following two primary factors: (1) The observed correlation between the percentage of repetitions performed and %VL in the set [20,74,75]. These results indicate that, against an equivalent %VL in the set, diverse players accomplish a similar proportion of repetitions relative to their maximal capacity before reaching muscular failure [20,74,75]. (2) A strong relationship exists among %VL in the set, fatigue levels, and metabolic stress encountered during different RT protocols [76]. Consequently, %VL in the set serves as a reliable indicator for assessing effort levels during a set. In summary, the optimal approach for prescribing, programming, and dosing training load during RT exercises appears to involve the velocity of the initial or best repetition of the set (reflecting relative load) and %VL in the set (indicative of volume) [76,77,78,79].

However, a common query arises regarding the absence of technology for measuring execution velocity in each training repetition. In such instances, these variables could be estimated through systematic coach observation, and training should be prescribed using the so-called level of effort, defined as the number of performed repetitions in relation to those to be completed against a given relative or absolute load. For example, if three sets of 5(8) repetitions are prescribed, it implies performing three sets of five repetitions with a load allowing approximately eight maximum repetitions. While this method has limitations concerning accurately determining the appropriate fatigue level for each individual, it serves as a viable alternative for prescribing training load while considering effort level. Table 3 illustrates the relationship between level of effort and %VL in the set. For the same relative load, a lower %VL in the set corresponds to lower effort levels. However, it is important to note that with the same %VL in the set, lower relative loads induce higher levels of fatigue [76]. Therefore, when training with low relative loads, reaching a lower %VL in the set is advisable to mitigate excessive fatigue levels.

In general, a set is considered to have a low level of effort if fewer than half of the possible repetitions are completed, a medium level of effort if half of the potential repetitions are attained, a high level of effort if more than half of the possible repetitions are completed, and a maximum level of effort if muscular failure is reached. An example of approaching the intended intensity through the level of effort is provided in Table 4, illustrating the full squat exercise.

## 6. What Exercises Should Be Used During Strength Training for Soccer Players?

The need of replicating specific technical movements during strength training (ST) to enhance performance gains is often emphasized. Nonetheless, a clear distinction exists between specificity and utility. Specificity refers to the degree of similarity between the trained exercise and the actual competitive gesture in terms of kinetics, kinematics, and joint range of motion, while utility pertains to the extent to which an exercise translates into improvements in sports performance (i.e., positive transfer). An exercise may exhibit specificity but lack utility if it fails to produce gains (i.e., positive transfer) in competitive actions. Conversely, another exercise may be deemed non-specific yet useful if it contributes to improved performance in competition. Currently, scientific evidence does not conclusively demonstrate that exercises emphasizing greater specificity lead to superior transfer to specific sport actions. Notably, a recent systematic review [21] and other studies [77,78,79,80] have indicated that training programs based on performing so-called basic exercises, such as back squats, offer substantial transfer effects on critical actions in soccer, including linear sprinting, vertical jumping, and directional changes. Moreover, similar findings were observed across studies involving soccer players of different age groups [22,23,55,56,57,58,59,60]. Therefore, the recommended approach for ST aimed at enhancing soccer players’ performance primarily emphasizes useful exercises, such as squats [8,46,47,51,81,82], combined with other exercises with greater specificity, such as unloaded jumping [83,84,85], loaded jumping [85,86], acceleration [87,88], Olympic weightlifting [89,90], or resisted exercises [91,92,93].

To operationalize the proposed physical training regimen encompassing fundamental movement patterns, six exercise options are delineated herein, suitable for ST in soccer players.

(a)Lower limb push: Squat or weighted sprint;(b)Lower limbs pull: Deadlift;(c)Horizontal upper limb push: Bench press;(d)Vertical upper limb push: Military press;(e)Horizontal upper limb pulls: Bench rows;(f)Vertical upper limb pulls: Pull-ups.

It is recommended that these exercises incorporate some type of “progression”, such as performing the same movements unilaterally. Likewise, it is advised that each exercise be practiced at a frequency of at least twice weekly.

## 7. What Combination of Volume and Relative Load Should Be Used for the Beginning of ST in Soccer?

In terms of load indicators or fatigue assessment, determining the minimum effective dose for performance enhancement is crucial. Several studies [59,77,78,79] have investigated the impact of training programs employing full squat exercises at different relative loads on athletic performance. These studies compared the effects of moderate-to-high relative loads (>60% of 1-RM, or MV lower than 1 m·s^−1^) with moderate-to-low relative loads (40–60% of 1-RM, or velocities between 1.28 and 1.00 m·s^−1^). They consistently found that training with lower relative loads led to greater improvements in high-speed actions such as linear sprinting, directional sprinting, and vertical jumping [59,77,78,79]. Furthermore, RT with lighter loads also resulted in enhanced resistance to strength or speed loss, as demonstrated by improved performance in repeated sprints with short recovery intervals [23,60]. This phenomenon could be attributed to the improvement in leg strength, wherein each action performed at the same absolute load (e.g., body mass) represents a lower relative load [78,79]. Additionally, recent data from our research group involving basketball players (unpublished) revealed a direct relationship between gains in the ability to repeat sprints and improvements in leg strength measured through full squat exercises after a full squat training program using relative loads ranging from 40 to 60% 1-RM.

In terms of volume, most studies [21,22,23,55,56,57,58,59,60] have used protocols comprising three training sets, performed twice a week, primarily focusing on the full squat exercise. These studies generally found that optimal effects on high-speed actions were achieved with small magnitudes of %VL in each set, typically ranging between 10 and 20% [21].

Based on published research findings [22,23,55,57,58,59,60], recommended relative loads for initiating RT in soccer players, irrespective of age and experience, involve moving at a MV close to 1.30 m·s^−1^ (~40% 1-RM) with %VL between 10 and 15% for the full squat exercise. Alternatively, initial training stages might involve lower relative loads (30–35% 1-RM or 1.40–1.45 m·s^−1^) with lower %VL in each set (5–10%). For upper limb exercises, slightly higher relative loads and %VL in each set may be allowed for greater strength gains.

## 8. Importance of Quantifying Training Load

The realm of physical training is susceptible to the emergence of “trends” or novel methodologies, often lacking scientific basis. Despite this, numerous exercises or protocols have garnered popularity within the broader training sphere, particularly among soccer players, fueled by endorsements from prominent physical trainers and athletes. One prevalent trend involves the integration of various “preventive” exercises, such as the Nordic curl [94], Copenhagen exercises [95], or those utilizing mini-bands or elastic bands [96]. Alongside the challenge associated with effectively controlling the load in these exercises, there are other important issues to consider when using these exercises. Firstly, the prevention of injuries is challenging as it often stems from multifactorial origins [97]. Secondly, it seems that one of the most effective procedures to prevent injuries is by enhancing strength and force production over time (i.e., improving RFD).

Undoubtedly, one of the pivotal roles of a coach and physical trainer in professional soccer is to monitor and tailor the training load (Figure 2) [20]. However, despite ongoing efforts, reliable tools and methodologies for quantifying the relative load or exertion level during training sessions for such exercises remain elusive. For instance, in the context of a Nordic curl exercise, two individuals may execute the eccentric phase at identical velocity. Nonetheless, variations in the strength levels of agonist muscles and participants’ respective body masses could lead to disparate relative loads. Consequently, physical trainers find it challenging to accurately determine the extent of load or fatigue imposed on their athletes, making it difficult to ascertain the cause behind specific outcomes.

Thus, as a general recommendation, exercises wherein the degree of fatigue cannot be meticulously controlled should not serve as primary exercises during RT. Without a clear understanding of the underlying causes, whether positive or negative, the precise degree of fatigue inducing changes in performance remains ambiguous. Consequently, the ability to pinpoint the factors influencing performance alterations becomes inherently compromised.

## 9. Practical Recommendations

From a practical standpoint, it is recommended for soccer players to perform the aforementioned exercises (full squat, deadlift, unloaded and loaded jump and sprint, bench press, bench rows, and pull-up exercises) at least twice weekly. During each session, they should aim for two to four sets with low perceived effort. It is crucial to execute each exercise with maximum intentional velocity (as fast as possible) during the concentric phase of the movement [98,99]. In the absence of velocity measurement systems, training might be prescribed based on effort level, weights, or relative loads that allow for more than 16 maximum repetitions, preferably even higher in the initial phases (i.e., loads that allow 25–30 maximum repetitions or more to be completed). Importantly, muscle failure should be avoided in each set [20,77]. Complete rest periods of at least 2–3 min between sets and exercises are recommended. Additionally, diversifying exercises within a session, including specific exercises without or with minimal additional load, such as jumping, accelerations, or directional changes, is beneficial [8,22,23,60]. However, exercises should not be combined in circuit form to prevent the accumulation of residual fatigue, particularly in high-velocity musculoskeletal actions. That is, it is advisable to finish all the planned sets of an exercise before moving on to the sets of the next exercise instead of performing alternating sets of different exercises.

A training program incorporating these principles has the potential to enhance performance in various critical soccer actions (e.g., linear and directional sprinting, vertical jumping, ability to repeat sprints, kicking ball speed), while also serving as a preventive measure against injuries, as RT was identified as an effective strategy for injury risk reduction [100]. Emphasizing full ranges of motion during exercises further promotes mobility and facilitates greater strength and transfer gains [101,102].

## 10. Conclusions

Velocity-based training (VBT) is not a training method, but a tool or procedure that has enabled a more precise quantification of training load. This accuracy has led to a better understanding of how to manipulate training load and volume to optimize high-speed actions in soccer players.

Evidence suggests that training with low loads (40–60% of 1RM) and low magnitudes of velocity loss (~10%) in the set during lower limbs exercise could be an effective strategy to enhance sprints, change in direction, and high-speed actions without generating excessive fatigue. This has reshaped strength training programs in soccer, shifting away from purely maximal load-based approaches and emphasizing the importance of movement velocity.

However, implementing VBT presents practical challenges, mainly related to the need for specialized technology to measure movement velocity. To overcome this barrier, alternatives such as the rating of perceived effort (RPE) have emerged as viable strategies when velocity-tracking devices are not available, allowing coaches to adjust training loads effectively based on the athlete’s perception and capacity in each session. However, these procedures are less objective and valid, and consequently, less useful for the precise control of training load.

In conclusion, VBT has contributed to an evolution in resistance training planning for soccer by providing tools to manage training load more precisely and in alignment with the demands of the game. Integrating these strategies into training could help improve performance, minimize injury risk, and optimize players’ physical preparation. Coaches and strength and conditioning professionals are encouraged to familiarize themselves with this new paradigm and adapt its application according to the available resources. 

## 11. Limitations

As a main limitation, we acknowledge that the intentional selection of articles may introduce bias in the interpretation of the results. However, this approach allows us to highlight key findings and discuss their applicability in the sports context.

## Figures and Tables

**Figure 1 jfmk-10-00106-f001:**
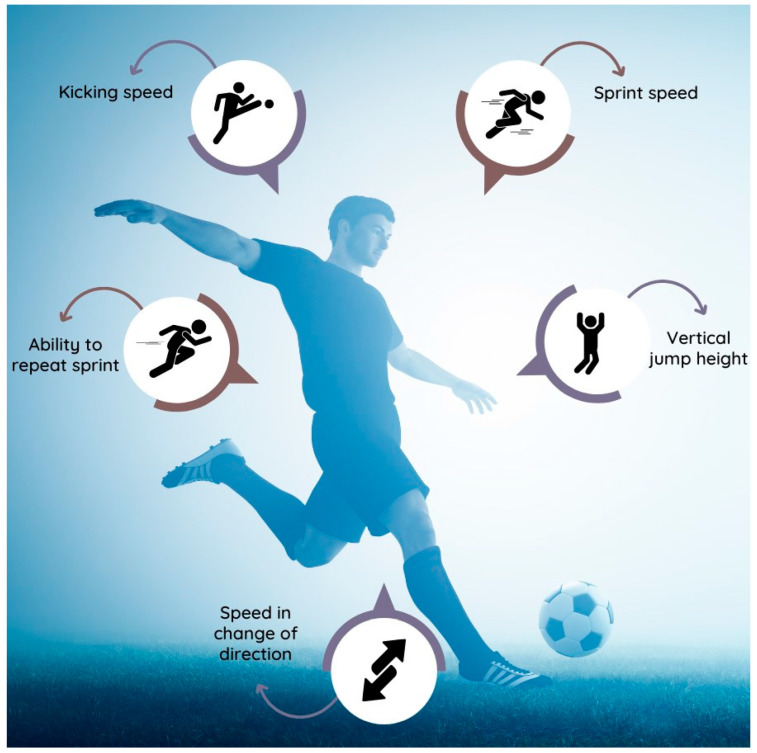
Primary objectives of physical preparation in soccer.

**Figure 2 jfmk-10-00106-f002:**
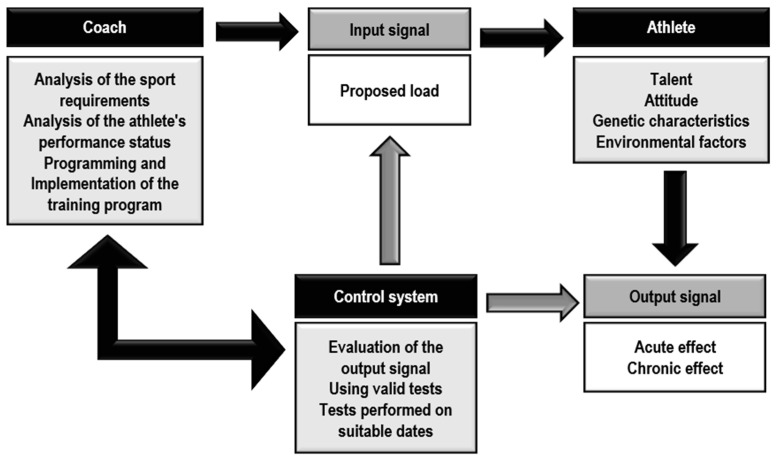
A schematic representation of the “Sports Training” system. It depicts the role of strength and conditioning trainers in professional soccer in controlling and dosing the training load and training effect as a fundamental basis for the development of training methodology.

**Table 1 jfmk-10-00106-t001:** Average distances covered by soccer players during matches across various leagues and competitions worldwide.

League/Competition Level (Sex)	Total Distance (m)
International Swedish/Danish (F)	10,000
Elite Swedish/Danish (F)	9700
Brazilian first division (M)	10,012
French Professional	11,000
Professional Australians (M)	10,100
Profession European Leagues (M)	11,393
Champions League matches (M)	11,010
Portuguese first division (M)	12,793
Elite Norwegian Juniors (M)	10,335
International Australian (F)	9140
Elite English (F)	12,400
Elite Danish (F)	10,300
Italian junior professionals (M)	9890
Japanese professionals (M)	10,460
Italian professionals (M)	10,860
Elite Danish (M)	10,330
English professionals (M)	10,659
European professionals (M)	10,864
Danish Premier League (M)	10,800
Swedish Premier League (M)	10,150
South American professionals (M)	8638
English professionals (M)	10,104
International English (F)	11,979
English professionals (M)	11,264
English Professionals (M)	10,274
U/19 Professionals (M)	9741
Champions League matches (M)	10,461
**Mean (SD)**	10,543 (904)
**Maximum**	12,793
**Minimum**	8638

M = male; F = female. Modified from Carling et al. [24].

**Table 2 jfmk-10-00106-t002:** Mean propulsive velocity associated with each relative load in different resistance exercises.

Load (% 1-RM)	Bench Press (m·s^−1^)	Pull ups (m·s^−1^)	Back Squat (m·s^−1^)	Bench Rowing (m·s^−1^)	Deadlift (m·s^−1^)
40	1.13	-	1.28	1.36	1.09
45	1.04	-	1.21	1.28	1.02
50	0.95	1.00	1.14	1.21	0.96
55	0.87	0.92	1.07	1.13	0.90
60	0.78	0.85	1.00	1.06	0.83
65	0.70	0.77	0.92	0.99	0.77
70	0.62	0.69	0.84	0.92	0.71
75	0.55	0.61	0.76	0.85	0.64
80	0.47	0.53	0.68	0.78	0.58
85	0.39	0.45	0.59	0.72	0.52
90	0.32	0.37	0.51	0.65	0.45
95	0.25	0.30	0.42	0.59	0.39
100	0.18	0.22	0.32	0.53	0.33

% 1-RM: percentage of one-repetition maximum. Data extracted from [19,20,68,69,70,71].

**Table 3 jfmk-10-00106-t003:** Relationship between level of effort and percentage of velocity loss.

Effort Character (EC)	Low	Medium	High	Maximal
Reps (MaxRep)	6(18)	6(12)	6(10)	6(6)
%VL	5–15%	20–25%	30–40%	>50%

Note: The values of percentage of velocity loss in the set (%VL) are indicative, as they depend on the relative load used. The values are presented for repetitions with a load that allows the subject to perform a certain number of maximum repetitions. The data are extracted from [21].

**Table 4 jfmk-10-00106-t004:** Determining volume and intensity through effort character in squat exercise.

Relative LOAD			Repetitions Completed at Different %VL in the Set			
% 1-RM	MPV (m·s^−1^)	Reps (Average)	10%	15%	20%	30%
50	1.14	23	5	7	8	11
55	1.07	19	5	6	7	8
60	1.00	16	4	4–5	6	7
65	0.92	13	3–4	4–5	5–6	6–7
70	0.84	10	3	3–4	4	5
75	0.76	8	2–3	2–3	3–4	4–5
80	0.68	6	2	2–3	3	4
85	0.59	5	1–2	2	2–3	3

% 1-RM: percentage of one-repetition maximum; MPV: mean propulsive velocity; Reps: maximum repetitions performed; %VL: percentage of velocity loss in set. Data extracted from [21,78,79].

## Data Availability

The data supporting this systematic review are from previously reported studies and datasets, which were cited.

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
