# Peer review of "Velocity-Based Training in Soccer: A Brief Narrative Review with Practical Recommendations"

_jfmk, 2025, doi:10.3390/jfmk10020106_

Round 1
Reviewer 1 Report
Comments and Suggestions for Authors
Velocity Based Resistance Training (VBT) in Soccer: A Brief Review with Practical Recommendations. REVIEW
This study aimed to review the scientific evidence on velocity-based resistance training to optimize key actions in soccer players.
The ease of reading and following the analysis, practical advice, which is acceptable to a wide professional audience, is the main strength of this work. On the other hand, the methodological setting of the work and its explanation represent the main flaw.
Although the title seems appropriate, I don't think there is any need for an abbreviation to appear in the title.
The first five rows of the abstract represent general knowledge; there is no need to spend so much space in the abstract on it.
Clear explanations of the goal as well as a detailed description of the methodological setting are missing in the paper. This needs to be tweaked.
First four rows of the text is also common knowledge; there is no need to spend so much space let along 15 references on it.
Rest of the text is very fair and good. It is easy to read, logically connected and provides a clear basis for inference and application. What is missing is a clear explanation of the goals and methodology of how the paper was written, which I think is important both for the readers and for the possibility of repeating the research or making an overall conclusion about it. Also, it is necessary to pay attention to the limitations of the study and explain how the transparency of this research would be increased. I ask the authors to consider whether it is justified to add all this to the text.
Reviewer 2 Report
Comments and Suggestions for Authors
The title is appropriate and captures attention - VBT relating to physical prep for soccer. Perhaps, to consider a small tweak to 'Velocity-Based Training (VBT)...", i.e., include the hyphen and omit 'Resistance' as VBT already implies resistance/strength training.
Abstract (Results): Directional (instead of direction) changes
Section 1 though comprehensive is a bit lengthy to illustrate the physical nature of soccer. Consider slight revision to keep it more succinct to improve the reading. The reader is kept waiting for your main point to come forth.
The flow of the subsequent Sections is good. However, I thought you should swap Sections 6 and 7, i.e., the Practical Recommendations should come in last.
The Reference list is also a bit lengthy for a 'Brief Review'. Consider trimming it as you revise Section 1 and perhaps the rest of the main text as well.
Added note for you to look at recently published paper in a related paper in IJSPP - Improving the Use of Lifting Velocity to Predict Repetitions to Failure: A Systematic Review by Miras-Moreno et al. 2025, 20, 335-344.
Reviewer 3 Report
Comments and Suggestions for Authors
Thank you for the opportunity to review this study. Here are my considerations about the study:
Abstract
1. I suggest reformulating the objective. It should have a clear verb and not "This review highlights";
2. Methods: According to the authors, a narrative review of the scientific literature was conducted. However, more clear information is needed, such as: number of studies included, databases searched.
Article structure
1. I suggest creating an Introduction section to show/present the State of the Art;
2. In its current form, the reader has no idea about the need for this study;
2.1 Therefore, it is essential to develop justifications for writing the study
3. Finally, the objective of the study should be presented
4. Immediately after the Introduction, the authors should present a section entitled Methodology. In this, the readers will be informed about the procedures adopted;
5. Authors should provide detailed information on the inclusion and exclusion criteria adopted, the number of studies, and the scientific bases researched.
6. After this, the results should be presented. These were very detailed.
7. I suggest that the section "7. Importance of quantifying training load" be presented before the section "6. Practical Recommendations";
8. In turn, authors should include a section entitled "Limitations of the study";
9. Finally, it is necessary to include a section entitled "Conclusion".
Reviewer 4 Report
Comments and Suggestions for Authors
See attached file.

Round 2
Reviewer 1 Report
Comments and Suggestions for Authors
I thank the authors for their cooperation, I can now support this version for publishing
Reviewer 3 Report
Comments and Suggestions for Authors
Dear authors, I believe that the study has been revised and is in better shape than before.
Minor correction
1. I suggest that section "2.1- Limitations of the study" be presented before section "8. Conclusion".
Reviewer 4 Report
Comments and Suggestions for Authors
See pdf. file
